# Using Community-Based Prevention Marketing to Generate Demand for Healthy Diets in Jordan

**DOI:** 10.3390/nu13093068

**Published:** 2021-08-31

**Authors:** Rowena K. Merritt, Jacqueline de Groot, Lama Almajali, Nitesh Patel

**Affiliations:** 1World Food Programme, Regional Bureau Cairo, Egypt & University of Kent, Canterbury CT2 7NZ, UK; 2World Food Programme, Jordan Country Office, Amman 11193, Jordan; jacqueline.degroot@wfp.org (J.d.G.); lama.almajali@wfp.org (L.A.); 3World Food Programme, Regional Bureau Cairo, Cairo 11728, Egypt; nitesh.patel@wfp.org

**Keywords:** community-based prevention marketing, social behavior change communications, Jordan, United Nations, malnutrition

## Abstract

Jordan has been experiencing a nutrition transition with high rates of micronutrient deficiencies and rising overweight and obesity rates. This highlights the need to generate demand for healthy diets. This study used a community-based prevention marketing approach and worked with local communities as partners to develop a set of behavior change interventions to improve healthy eating within vulnerable communities. Individual, family, and paired-friendship interviews, and co-creation workshops were conducted with 120 people. The aim of these interviews was to gain an in-depth understand of school-aged children and their families’ nutrition knowledge, attitudes, and practices, including social and cultural norms and behavioral determinants, and then use this information to co-create interventions, activities and materials targeted at supporting school-aged child nutrition. Analysis of the interviews revealed that dietary habits are both deeply personal and profoundly entwined by emotions and social norms, and that parents often gave in to their children’s demands for unhealthy foods and beverages due to their perception of what a ‘good parent’ looks like and the desire to see their child ‘smile’. These key insights were then shared during the co-creation workshops to develop behavior change interventions—ensuring that interventions were developed by the community, for the community.

## 1. Introduction

The Middle East and North Africa (MENA) region is undergoing a nutrition transition [1]. The coexistence of food insecurity, undernutrition, alongside overweight and obesity is a growing challenge across the region [2]. Overweight and obesity are increasing quickly due to rapid shifts in diet and activity levels [3]. Additionally, the region has been affected with on-going conflicts for many years, further affecting the nutrition status of the most vulnerable population including women and children [4].

### 1.1. Prevalence of the Double Burden of Malnutrition in Jordan

Jordan is a small, upper-middle-income country with few natural resources and water scarcity [5]. It has a population of nearly 11 million, of which 2.9 million are non-citizens, including refugees. The population has grown rapidly over the past decade, in part due to the Syrian crisis which started in 2011. In Jordan alone, the influx of Syrian refugees has increased by almost ten-fold since July 2012, totaling over 650,000 registered refugees registered [6]. 83 percent of the refugees in Jordan reside outside the refugee camps; renting their own apartments, staying with host families or living in informal settlements [7]. The remaining percentage living in camps.

In Jordan, the diet is mainly based on wheat, rice, vegetables, and foods of animal origin (milk/dairy products, and meat). Dietary diversification has improved and is currently quite high, however there has been an increase in the consumption of vegetable oils and sweetened, processed foods and drinks in recent years [5]. As a result, Jordan now faces a double burden of malnutrition, marked by the persistence of micronutrient deficiencies and the increase in overweight and obesity. The National Micronutrient Survey (2010) revealed some concerning conclusions regarding the nutrition situation of the population—especially women [8]. Prevalence of anemia for non-pregnant women was 30.6 percent. Older women and married women were more likely than younger unmarried women to have anemia and iron deficiency anemia. Prevalence of anemia for children was 17 percent. Vitamin D deficiency was prevalent in 60.3 percent of non-pregnant women. Prevalence of vitamin D deficiency among children was nearly 20 percent (19.8%). Obesity rates are also increasing, with a more recent cross-sectional study conducted in 2015 finding 17.3 percent of the children were overweight, and 15.7 percent obese [9].

### 1.2. The Role of Demand-Creation for Healthy Foods

Many programs run by international and national organizations focus on the supply of nutritious food products and fortified foods, for example, oil fortified with vitamin A and vitamin D, or on the provision of supplements, such as iron tablets to reduce rates of anemia. Whilst both these elements are important, and food fortification has proven an effective and low-cost way to reduce micronutrient deficiencies [10], the general availability of cheap, high added sugar food and drinks has contributed to a growing obesity rate [11]. Few programs focus on generating demand for healthy foods or reducing demand for unhealthy foods, and whilst stunting rates have reduced, rates of micronutrient deficiencies and obesity continue to rise [8].

There is a growing body of evidence which indicates that higher consumption of sugars and foods and beverages containing added sugar, is associated with a greater risk of dental caries, type 2 diabetes mellitus and increases in body mass index [12,13]. In 2015, the World Health Organization recommended that everyone—children and adults—should limit their daily sugar intake to less than 10 percent of all calories consumed. For children, this equates to about 45 g of sugar a day [14]. Guidelines from other countries suggest that people, in particular children, should have even less sugar [15]. However, it is not always easy to determine how much sugar is in foods and beverages, and whilst most parents know that too much sugar is unhealthy, they are often less aware of how much sugar is in popular snack foods and drinks, or how much sugar a child should be consuming each day [16].

Focusing on the supply of healthy foods and raising awareness about the consequences of unhealthy eating practices, or the dangers of not eating enough of certain healthy foods is not enough to make people eat a diet that meets recommendations for health. There is a growing body of evidence from the field of behavioral science which challenges the assumption that people are rationale decision-makers and that people do not make decisions around food choices and nutrition in isolation or based solely on perceived health benefits [17]. Instead, research shows us that we need to move beyond supply and information giving; we need to amplify and scale-up at speed innovative solutions that use the power of emotional appeals, social influences, and choice architecture (the design of different ways in which choices can be presented to consumers, and the impact of that presentation on consumer decision-making). One way to do this is to use a community-based prevention marketing (CBPM) approach [18]. 

### 1.3. Community-Based Prevention Marketing

CBPM is a community-directed social change process that applies social marketing theories and techniques to the design, implementation, and evaluation of health promotion and disease-prevention programs [19]. It is about working with local communities as equal partners and active participants, thereby spurring action by a community, for that community. 

A core feature of CBPM is its focus on listening to the wants, needs, desires and values of the individual and the communities, as opposed to simply looking at people’s needs from a clinical or nutritional standpoint. At the start of the CBPM process, formative research to listen to the community is conducted, with the aim of understanding what could motivate behavior change, as well as the barriers to change—physical and psychological barriers and perceived and actual ones. 

Insights gained from the formative research are combined with social marketing concepts to develop effective interventions to facilitate, support and motivate behavior change. By taking a social marketing:
a conceptual framework is used to guide intervention development consistent with exchange theory. Exchange theory is fundamental to social marketing; if you want people to change their behavior, you have to offer them something in return. The target audience must perceive benefits as equal or exceeding perceived costs [20]. The benefits are often non-monitory—a feeling of belonging, sophistication, or security, etc. Or an alternative product that gives them the same or greater benefits as the product they are currently using. These benefits have to be short term as people value these more than longer term ones.formative research is conducted which seeks input from the target population and to gain understanding around people’s aspirations, values, fears, and perceptions of the intervention’s perceived benefits versus perceived costs [21].a range of interventions at the individual and environmental levels (as opposed to just focusing on advertising and communications) are developed, considering all four levers of change [22]:
○Products or services to support the desired behavior change.○Changes to the physical environment (design) to support a change in behavior.○Development of messages and materials to meet the information and educational needs of the target audience in relation to the desired behavior change.○Control mechanisms such to either incentivize the desired behavior, or disincentivize it, e.g., sugar-taxes.

This paper details the findings from formative research conducted in Jordan and describes how co-creation was then used to develop a comprehensive behavior change strategy and intervention mix aimed at school-aged children and their parents to improve child nutrition. The research included both parents, grandparents, and children from the Zaatari camp—a large Syrian refugee camp—as well as Syrian living in the community and Jordanian families. The work was conducted by the UN World Food Programme (WFP) as part of its overall strategy to reduce food insecurity and improve nutritional outcomes in Jordan.

## 2. Formative Research: Materials and Methods

The flow of the formative research and intervention development stages are presented in Figure 1. 

First, a review of the literature was done to identify any potential knowledge gaps in relation to the topic are and the target audience of Jordanian and Syrian families living in Jordan. Second, a mix of individual, family and paired-friendship interviews were conducted. The paired-friendship interviews were only conducted with the children, and the children would select one other friend who they wanted to be interviewed with (i.e., the participants determined who their friends were, as opposed to the researchers or a teacher/parent).

All the interviews were qualitative in nature and were conducted with the aim of generating an in-depth understanding of school-aged children and their families’ nutrition knowledge, attitudes, and practices, including an understanding of the social dynamics around eating preferences and habits within households.

Third, after the findings of the interviews had been analyzed, co-creation workshops were conducted. Co-creation refers to an intervention or campaign design process in which input from consumers plays a central role from beginning to end. The aims of the workshops were to co-design interventions, activities and materials targeted at supporting school-aged child nutrition.

Table 1 details the type and number of participants involved.

### 2.1. Individual, Family and Friendship Interviews: Data Collection Methods and Participants

To help inform and create a structure for the discussions, a guide was initially developed based on the findings from a literature review of articles published on healthy eating and dietary choices. The main topics for discussion were: (a) school-aged children and their families’ nutrition knowledge, attitudes and practices; (b) social and cultural norms that influence children’s nutrition-related behaviors; (c) behavioral determinants of optimal child nutrition, including the barriers and motivators to optimal child nutrition.

The data collection and analysis followed an iterative process, whereby emergent themes from early interviews were used to add to or refine questions during subsequent discussions. All interviews were conducted during 2018 and 2019 by local external researchers who had training in and experience of conducting qualitative research in Joran. None of the interviews were conducted by WFP staff to avoid potential bias. The participants were selected using purposive sampling, meaning they were selected because they possessed knowledge that was directly related to the research questions [23]. Sampling considered age, gender, location, nationalities (Jordanian/Syrian), and living situation (living in the community/refugee camp). WFP used a combination of snowball sampling and recruited through local community groups and existing WFP activities, such as the Healthy Kitchen project. The formative data was collected in three locations: Zaatari camp, Ein el Basha, and Karak. Located in north Jordan, Zaatari is the largest refugee camp in the Middle East and North Africa region currently hosting more than 76,000 Syrian refugees coming from Dara’a district in southern Syria. Zaatari camp and Ein el Basha were chosen as they are vulnerable poverty pockets where WFP undertakes other activities, most notably the school feeding program, and because both Syrian refugees and Jordanians live there. Karak was added to increase geographical spread of the sample. All of the participants had at least one child of primary school age (aged between 5–11 years old) and 25% of the participants were Syrian. 84% were female and the age range of participant was 8 to 71 years old.

In order for participants to feel at ease during the interview, most of the participants were offered the opportunity to be interviewed in their own home or in a location that was convenient to them. All participants were interviewed once and interviews took between 30 min to 1.5 h, with an average length of 55 min.

### 2.2. Research Questions

For this study semi-structured interviews were conducted. At the start of each interview, loosely structured, open-ended questions were asked. In order to pursue an idea or response, more detailed questions were subsequently asked, or prompts made. The wording was not standardised, as the interviewers tried to use the participant’s own vocabulary when framing supplementary questions.

The questions covered the following areas:current knowledge, attitudes and practices of school-aged children and their families relating to nutritionthe knowledge, attitudes and practices of school-aged children and their families’ relating to health and health concernsSocial and cultural norms which influence children’s nutrition, including appropriate strategies to influence positive social change and create new social normsbarriers and motivators to optimal child nutritioninfluences school-children’s health and nutrition behaviorsappropriate channels, entry points and existing service delivery platforms to reach the target audienceskey influencers at the primary, secondary and tertiary levelstrusted individuals, organizations, and sources of information

The guide was used as an ‘aide-memoire’ and as a general framework for discussion, ensuring that all themes were covered with the necessary prompts but, at the same time, enabling discussions to be spontaneous, flexible and responsive to the thoughts and opinions of those being interviewed.

### 2.3. Data Analysis

All interviews were audio recorded with permission from the participants and transcribed verbatim, then translated into English for analysis purposes. The translations were checked by the national project lead for accuracy before being analyzed. Transcriptions were imported into NVivo [24], and analysis followed a thematic approach to identify key themes and codes [25]. Two researchers from different disciplines analyzed the first five interviews and compared coding. One researcher then went on to analyze the subsequent research data due to the consistency of the initial coding. Data collection and analysis continued until saturation occurred (i.e., the point at which no new significant themes emerged).

No incentives were offered to participants to take part in the interviews.

## 3. Formative Research: Results

Analysis of the data revealed three broad themes around: (1) current eating preferences, habits and behaviors; (2) social and cultural norms influencing nutrition; and (3) barriers and enablers to improved nutrition.

### 3.1. Current Eating Preferences, Habits and Behaviors

In general, there was good awareness around what a healthy diet included (and what it did not include). Whilst children were unconcerned with their dietary habits, parents worried that their children ate too many unhealthy, sugary snacks. “*Chips and biscuits*” were the most frequently mentioned snack foods. Despite this concern, all enjoyed “*treating*” children with sugary snacks and used them as a way to show their love.


*“I cannot describe to you my feelings. I love them [grandchildren] very much, I don’t bother them, and I buy them everything, especially ice-cream.”*

*(grandmother)*


Most of the parents and children consumed soft drinks frequently (usually once a day, but sometimes up to three times per day). They stated they were “*addicted*” and therefore needed to consume such beverages daily to “*feel comforted*”.


*“She [the daughter] feels comforted when she drinks it, the same like coffee addicts…It relieves her headache. It has nothing to do with the price. It has become a habit.”*

*(mother)*


Despite parents and grandmothers expressing concern over the number of soft drinks the children drank, these concerns rarely deterred children from drinking soft drinks. This left parents feeling as if they had no control over their children’s dietary choices.


*“My son fills a big glass with Pepsi although he knows its harm from the Internet. He never listens to me, he does this every day for a week.”*

*(mother)*


The parents interviewed from the refugee camp often talked about their inability to plan due to their current ‘in transit’ status. This often created a ‘live for the moment’ mentality, resulting in them prioritizing their children’s immediate happiness over long-term health risks. As a result, they often bought sweets and other sugary snacks and drinks for their children as it “*made them smile*”.

### 3.2. Social and Cultural Norms Influencing Nutrition

The grandmothers often lived with the parents and children, and either decided what should be cooked or did the shopping for the whole family. Everyone ate the same meals; however, the children had a lot of control over what was cooked. The other family members would accommodate their wishes as they believed it was better for the children to have food in their stomach, as opposed to ‘going to bed hungry’.


*“She doesn’t eat if we don’t bring her what she wants.”*

*(father)*



*“Yes, she says ‘I don’t want to eat from your food, you either bring me fast food or I sleep.’ Sometimes I pity her and bring her fast food.”*

*(father)*


Parents, in particular the mothers, were very worried about other people’s opinions within the local community, and they believed if they said ‘no’ to their child and then their child “*made a bad scene*”, then others would question their ability as a mother. This meant that most parents ‘gave in’ to their children’s demands for unhealthy foods and sugary beverages.


*“I feel embarrassed when they keep crying in shops or when we are on a visit to some acquaintance.”*

*(mother)*


People perceived a healthy child as being “*chubby*”. As one grandmother responded when asked what a healthy child looks like:


*“Someone who is knowledgeable, with a chubby face, correct speech and an open mind.”*

*(grandmother)*


### 3.3. Barriers to Improved Nutrition

#### 3.3.1. Pester Power

Most of the parents brought up ‘pester power’ as a major barrier to their children eating healthy. They said their children would often ask for certain foods (e.g., chips, candy, soft drinks). Parents gave in to stop the child from “*nagging*” or crying, especially if in public.


*“I give them Pepsi because they nag a lot.”*

*(mother)*



*“Yes, I bought them a pack of lollipops because if I don’t buy them these things they keep moaning.”*

*(father)*


#### 3.3.2. Competing Priorities

For most of the mothers, the main concern was ensuring that their children achieved high school grades. This was a concern for fathers also; however, it was the mothers who followed up with their children’s daily studies. Those living in the refugee camps were also concerned about the safety of their children at night. Due to these concerns, although they hoped their children would eat healthily, it was a low priority.

#### 3.3.3. Decreased Appetite and Fatigue

Many of the mothers discussed decreased appetite and fatigue as reasons why they felt their children were not eating enough or eating healthy foods. This was often a result of the children not being hungry at mealtimes as they snacked on “*chips and biscuits*” throughout the day.


*“I have problems with [child’s name] about food, I even run after her with food…it isn’t easy. She only wants chips and biscuits.”*

*(mother)*


#### 3.3.4. Perception of Who Eats Healthy

A few of the parents said they sometimes could not afford to feed their families healthy foods. They perceived only wealthy people as being able to afford such “*luxuries*”.


*“They are people who have money to buy everything they want and eat fruits all the time, like the king and the doctor, they have everything.”*

*(grandmother)*


However, a few of the mothers disputed the financial aspect.


*“It is not financial; they simply prefer to snack on junk.”*

*(mother)*


### 3.4. Possible Enablers to Improved Nutrition

#### 3.4.1. Future Aspirations

Whilst the parents and grandmothers rarely had any personal aspirations, they projected their hopes and dreams onto their children. Most of the parents, unpromoted, said they wanted their children to receive a good education, achieve high grades, and find a “*good job*”.

However, when asked what would help the children achieve these dreams, healthy eating was never mentioned by the participants. Instead, lack of money and a poor environment were mentioned.

#### 3.4.2. Being a Good Parent

All family members talked with pride about their children and the parents explained how they tried always to be “*good parents*” by putting their children’s needs and wants first. Being viewed by others within their community as being a good parent was also important to them.

## 4. Development of Intervention Ideas through Co-Creation Sessions

After the formative research had been conducted and the data analysis had been finalized, the findings from the interviews were shared and discussed with the people attending the co-creation sessions. Six co-creation sessions were run by an external facilitator and involved local stakeholders and mothers and grandmothers of school-aged children. Only female family members were invited to attend the co-creation sessions due to gender sensitivities. The aims of the co-creation sessions were to build trust, buy-in and feedback in the design, planning and implementation of the behavior change interventions. As with the formative research, the co-creation attendees were recruited through local community groups and existing WFP activities.

As the initial interviews had highlighted the vulnerability and feelings many of the parents felt in relation to their current situation, the WFP team recognized that the families might struggle to find the right words to express how they are feeling and their true values and beliefs. To avoid the well-documented risk of people simply saying what they think people want to hear [26], a slightly different approach was taken to the co-creation sessions. Before the sessions, mothers were asked to take photos or draw images to explore the following questions:“What pictures express your challenges everyday in getting your children to eat healthier food?”“What does a good mother look like to you?”“What makes you smile?”

The sessions were divided into three parts. During the first part, the photos and drawings taken by the attendees were discussed, as well as the findings from the formative research, to help gain a unique understanding of the world through the women’s eyes. The second part of the sessions focused on the development of possible interventions and discussed the key elements that the interventions should include. For example, the women believed in the importance of family cooperation, and helping each other, and that any interventions should promote and support them to spend more “*quality time*” with their children. The final part of the sessions focused on the development of a campaign and working with the women to develop message and imagery ideas.

The co-creation sessions were conducted in 2019. Ten participants attended each of the six sessions.

### Development of a Behavior Change Strategy

The feedback from the co-creation sessions, as well as discussions and input from the Ministry of Education and the Ministry of Health, and consultations with stakeholders were all used to develop a final social behavior change strategy and campaign plan. The aim of the work is to reduce the consumption of high sugar beverages and high sugar snack foods among school-aged children. The interventions include:

1.Developing a social marketing campaign to promote easy, realistic choices and changes.2.Taking a whole family approach through family pledges and workshops.3.Partnerships with retailers to increase stock and consistency of supply of healthier options, including nudge marketing activities in stores (for example, putting fruit and healthy snacks right beside checkouts instead of high-sugar snacks and putting healthier alternatives in the center of the display, and unhealthy ones higher up or lower down the display) and the promotion of ‘sugar swaps.’ Sugar swaps is about encouraging people to swap high sugar products for low sugar alternatives, for example, swapping flavored yogurt for Greek yogurt with fresh fruit.4.Control mechanisms to create new social norms around healthy eating, including a sugar tax and clear food labelling.

The social marketing campaign was developed to create as sense of need and challenge exiting social norms and the perception that a “chubby” was a healthy child. The campaign is designed to be implemented in two stages due to the current content with existing diets and the lack of perceived need to change identified through the formative research.

Stage one focuses on creating the right preconditions for behavior change, creating concern that too much sugar consumption has health consequences for their children and that too much sugar can have short-term concentration issues (affecting academic ambitions). Stage two is developed to support people on the behavior change journey. For people to move from intent to actually changing their behavior, they need to be convinced that change is possible and normal. Therefore, stage two aims to inspire people to believe that change is possible and convince them that change is already happening. Stage two of the campaign also focuses on sugar swaps, which can be built into people’s existing habits, as opposed to banning certain foods. The research found that parents felt a lack of control and an inability to influence their children’s diet habits, as well as feeling guilty for the current situation. Therefore, this approach was more supported during the co-creation events as family members feared it was too difficult to say ‘no’ to everything. The campaigns call to action is:


*‘‘No’ Means ‘Yes’. Motherhood for Life.’*


The slogan emphasizes that by saying ‘no’ now, you are saying ‘yes’ to a whole future of positive activities and outcomes for your children, and that by giving sweet foods and drinks to children now, mothers are creating habits which will lead to a lifetime of ill health for their children.

The interventions, strategy and campaign plan were finalized in 2020, and is due to be implemented in late 2021 when schools are expected to reopen after a long period of school closures due to the COVID-19 pandemic.

## 5. Discussion

It has been well evidenced that knowledge is often not enough to achieve sustainable healthy behavior changes [27], and this research highlights this issue as despite knowing what they should (and should not eat), people continued to make unhealthy choices. The research also highlights that dietary habits and tastes are both deeply personal and profoundly entwined by emotions and social norms. This finding is consistent with other studies which found that social and cultural factors, food beliefs and perceptions influenced dietary behavior among refugees and vulnerable groups [28,29,30]. This means that people do not make decisions around food choices and nutrition in isolation or based solely on health outcomes or perceived health benefits.

The issue of ‘present-bias’ [31] also came clearly from the research, highlighting that people often prioritize the immediate happiness of their children as opposed to thinking about the longer-term health risks. This is not a unique issue to Jordan; globally parents greatly value children’s happiness, citing it well above other possible priorities [32], and a happy child is very much linked to perceptions of good parenting [33]. However, it is perhaps arguable that this concept of present-bias is particularly relevant to vulnerable population groups such as refugees. This research found that the parents often felt guilty about their living situation and try to compensate by giving their children unhealthy foods to make them ‘smile’. This suggests that any health promotion activities which focus on the longer-term health benefits may fail to resonate or result in positive change.

By using a CBPM approach, the WFP team were able to understand the issue of healthy eating from the audience perspective and explore how the emotional, social, and behavioral dynamics influence health eating practices and decision-making. Through the co-creation process, a sense of ownership and commitment to change was developed within the most vulnerable communities, and the use of pictures and drawings helped people to express their feelings. Other studies have also identified the benefits of using such approach, in particular when working with vulnerable and marginalized population groups [34]. With this approach, the participants become co-owners of the solution, and this also may increase the chance of maintaining the participation of the target group in future activities [35].

The questions asked in this study were also important in exploring the dynamics and aided the development of targeted interventions and messages. Often research focuses on dietary questions, inquiring what people eat, and the quantity and frequency. However, this fails to gain an understanding of *why* people make the food choices they do [36]. Instead, by focusing on qualitative interviews and using co-creation methods, instead of more structured surveys, you can understand, not just what was consumed, but also why it was consumed, how they felt before and after [37]. By asking these broader questions, this project gained an understanding of how parents feel when their child asks for unhealthy foods, and the internal dialog they had with themselves over whether to give them the unhealthy food and beverages or not, and how they felt after (for example, pleased that their child was now happy, feelings of guilt, etc.).

In addition to this, by asking people general questions such as—‘what makes you smile’—key insights can be gained to possible motivators for change and help practitioners and communicators move away from the traditional health focused messages which are frequently used.

## 6. Conclusions

CBPM is a participatory approach to social change that is distinguished from other community-based approaches by the use of a social marketing mindset to explore and solve public health problems [18]. This project shows that by asking more holistic questions you can gain key insights that take you beyond simply looking at the issue from a nutritional and health standpoint. By treating the communities as active and equal partners, this can result in developing a mix of interventions which overcome the barriers to change and importantly, motivate change by communicating and providing benefits that the target audience care about, such as the perceived psychological benefits of feeling like a good mother and helping your child achieve their dreams.

## Figures and Tables

**Figure 1 nutrients-13-03068-f001:**
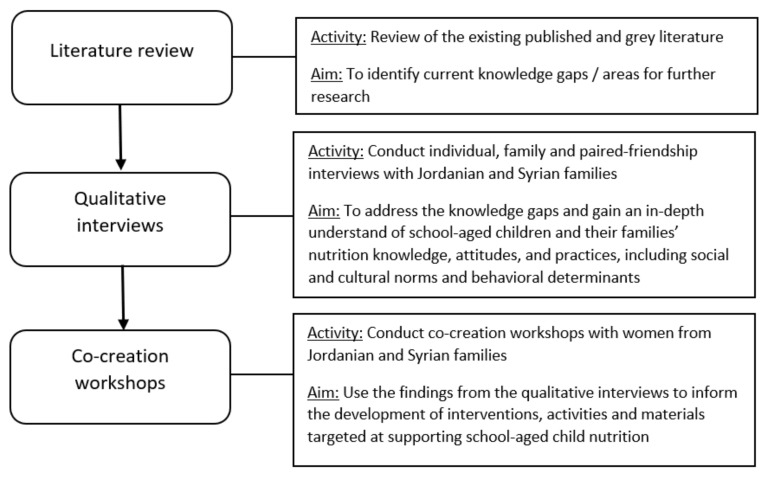
Process used.

**Table 1 nutrients-13-03068-t001:** Number of respondents from the different target audiences.

Audience	Number Engaged with
Mothers	89
Fathers	16
Grandmothers	9
Children	6

## Data Availability

The data presented in this study are available on request from the corresponding author. The data are not publicly available due to ethical restrictions.

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
