# Peer review of "Using Community-Based Prevention Marketing to Generate Demand for Healthy Diets in Jordan"

_nutrients, 2021, doi:10.3390/nu13093068_

Round 1

Reviewer 1 Report

The article presents a program of community intervention in a refugee population in Jordan using a method of participation in social marketing. 

I suggest the authors to answer the following questions with a maximum of 200 characters:

  1. what is known about the topic? 
  2. what does the study add to the literature? 
  3. Which are the implications of the results of this study? This means the practical implications of the results for policy, public health and health management, in order to reduce the gap between research and practice.

ABSTRACT

The objective is not clear in the abstract. 

The target population of the intervention is not included. 

The results do not show the impact of the interventions achieved. 

INTRODUCTION

- The authors base the justification of the study on the general results of a Jordanian National Micronutrient Survey from 10 (2010) years ago.   Question: Are there more recent data? Does the same epidemiological profile exist, or are there changes?

- The document describes worrying data on the nutritional status of the population, such as increasing obesity, anaemia and vitamin D deficiency. The responses of international and national organizations focused on preventive measures to provide nutritious and other fortified foods.

What impact have these preventive measures had on solving the identified health problems? Have the changes been achieved?

- The introduction does not include qualitative studies of community nutrition interventions and their impact on solving nutritional problems. 

- Lines 120 to 123: The objective is not clear, and it is not specified how long it is expected to change behaviors. Nor does it include the specific refugee population.

METHODOLOGY:

- It is difficult to follow the reading of the methodology. An organized methodology would be easily reproducible.

- I suggest the authors make an effort to organize the information in stages. Each stage should include a brief description, the actors and the time each stage takes. A process diagram would help to improve the methodology.

-The authors start directly with the technique of collecting information through interviews. It would be advisable to begin with the type of community intervention to be carried out, taking into account some stages according to the review.

RESULTS

It is well-structured, however, it is dense to read. I suggest an outline of the main results of the study.

DISCUSSION

There is no comparison of its results with other studies of similar or different interventions.  This point is very important.

REFERENCES  

  1. Of 28 references cited, 39% (n=11) correspond to scientific literature and the rest to gray literature. 
  2. Review the way of citing. In the references, there is NO homogeneity in the way of including the years of publication, names of the journals, and titles.

In relation to the gray literature, it is also described in different forms. The pages consulted should be included. 

3.There are some incomplete references. 

4.The bibliographic references constitute the basis of the evidence of the study and the contribution to science. 

Author Response

Thank you for your input and comments. I have addressed the points in bold individually below.

Many thanks

ABSTRACT

The objective is not clear in the abstract. Added in “The aim of these interviews was to gain an in-depth understand of school-aged children and their families’ nutrition knowledge, attitudes, and practices, including social and cultural norms and behavioral determinants, and then use this information to co-create interventions, activities and materials targeted at supporting school-aged child nutrition.”

The target population of the intervention is not included. Added/detailed in the objectives  

The results do not show the impact of the interventions achieved. This paper focuses solely on the formative development stage of the work and the findings from the formative research, and does not focus on the results of the project as these have been delayed dure to schools being closed in Jordan for over a year.

INTRODUCTION

- The authors base the justification of the study on the general results of a Jordanian National Micronutrient Survey from 10 (2010) years ago.   Question: Are there more recent data? Does the same epidemiological profile exist, or are there changes? A 2015 study has been included already and that was the most un-to-date published research we identified. There is more recent data as a new survey was done a few years ago by the Government and the UN. But the Government have still not approved the findings and therefore we are not permitted to use it. This data shows the situation is worsening in relation to obesity and overweight.

- The document describes worrying data on the nutritional status of the population, such as increasing obesity, anaemia and vitamin D deficiency. The responses of international and national organizations focused on preventive measures to provide nutritious and other fortified foods. I have added in a sentence explaining that programs do not focus on generating demand (they are supply focused) and they fail to have the impact needed.

- What impact have these preventive measures had on solving the identified health problems? Have the changes been achieved? See addition above.

- Lines 120 to 123: The objective is not clear, and it is not specified how long it is expected to change behaviors. Nor does it include the specific refugee population. I have added in the overarching objective and also the target audience. However, as this paper focuses on the development of the intervention as opposed to the outcomes of the intervention, the time to change has not been discussed (and due to the schools being closed for over a year creating implementation delays, as well as the dynamic nature of behaviour, it is hard to put a time on it).

 METHODOLOGY:

- It is difficult to follow the reading of the methodology. An organized methodology would be easily reproducible. Sub-headings added and some restructuring has been done.

- I suggest the authors make an effort to organize the information in stages. Each stage should include a brief description, the actors and the time each stage takes. A process diagram would help to improve the methodology. Diagram added (Figure 1).

-The authors start directly with the technique of collecting information through interviews. It would be advisable to begin with the type of community intervention to be carried out, taking into account some stages according to the review. Restructured now.

 RESULTS

It is well-structured, however, it is dense to read. I suggest an outline of the main results of the study. I have tried to reduce slightly so less dense now.

 DISCUSSION

There is no comparison of its results with other studies of similar or different interventions.  This point is very important. Comparisons now discussed.

 REFERENCES  

  1. Of 28 references cited, 39% (n=11) correspond to scientific literature and the rest to gray literature. 
  2. Review the way of citing. In the references, there is NO homogeneity in the way of including the years of publication, names of the journals, and titles.

In relation to the gray literature, it is also described in different forms. The pages consulted should be included. 

3.There are some incomplete references. 

4.The bibliographic references constitute the basis of the evidence of the study and the contribution to science. 

Many of the references referring to nutrition rates in the MENA region and Jordan come from UN and government organisations so are grey literature. Even when talked about in academic papers, they refer to the same studies conducted by the UN / government. However, I have tried to add in a few more academic papers and also formatted the references so they are in style of the journal.

Reviewer 2 Report

This manuscript describes the formative research and development of strategies and messaging for a social marketing campaign to improve child nutrition in Jordan. My opinion is that the paper has the potential to be of strong interest to the readers of this journal, given the topic, target audience, timeliness of the work, and community-based participatory approach.  Lack of important methodological details make it difficult for me to judge the scientific merit, but I think the authors should be able to respond and improve the methods section. The formatting of the paper is off in some places, but again the authors should be able to fix all the format errors. I think the findings are interesting and mirror what other researchers have reported in developed countries among immigrant populations who experienced poverty and other hardship in their own countries. I would strongly encourage the others to delve into that body of work and explore the parallels with their own research. This would greatly strengthen the discussion and overall value of the paper. Specific comments:

  • Line 70: I think the recommendations are for added sugars and suggest a clarification
  • Line 120 and 134: It appears that the main focus of the campaign is to be on child feeding practices but that is not clearly stated in the purpose (line 120) or even the abstract. This focus should be clear in both the abstract and purpose, as well as the key words. Also, it seems that the target audience was mainly from refugees but that is also unclear.
  • Line 121: Co-creation is mentioned but I suggest a clear definition. I figured out later what was meant but think it’s helpful for an international audience to define terms in the intro. Also, what is meant by “choice architecture” (line 84)
  • Lines 108-119. Fix the formatting here. These are fragments or phrases, not sentences. Also fix lines 100, 105. Typo on line 41
  • Methods section: I have multiple questions/comments here. What was the training or expertise of the staff who conducted the interviews? When transcripts were translated into English, did more than one researcher verify accuracy of the translation? Also, did more than one researcher compare their codes and themes and how were the data merged? This is important to establish validity. Did nutrition program managers, known to the participants, conduct some of the interviews? If yes, what effect might this have had on bias in parent reporting concern about too much sugar intake (for example). I think readers really should be able to see the full questioning guide and suggest the authors make that easily available to readers. In line 125, what was a friendship interview? Also, if families were interviewed, how did that influence response patterns? Were there incentives offered for the interviews? How long did interviews take?
  • Sample characteristics: Can the authors provide any more details on the characteristics of the sample (beyond Table 1), like educational level of parents, age of children.
  • Line 309: Define terms “nudge marketing”, “sugar swaps”
  • Discussion: Compare your findings to those of other researchers who have reported on child feeding practices in immigrant populations.

Author Response

Thank you for your input and comments. I have addressed the points in bold individually below.

Many thanks

This manuscript describes the formative research and development of strategies and messaging for a social marketing campaign to improve child nutrition in Jordan. My opinion is that the paper has the potential to be of strong interest to the readers of this journal, given the topic, target audience, timeliness of the work, and community-based participatory approach.  Lack of important methodological details make it difficult for me to judge the scientific merit, but I think the authors should be able to respond and improve the methods section. The formatting of the paper is off in some places, but again the authors should be able to fix all the format errors. I think the findings are interesting and mirror what other researchers have reported in developed countries among immigrant populations who experienced poverty and other hardship in their own countries. I would strongly encourage the others to delve into that body of work and explore the parallels with their own research. This would greatly strengthen the discussion and overall value of the paper. Specific comments:

The methods section has been changed a lot to add in the missing information. It now reads:

FORMATIVE RESEARCH: MATERIALS AND METHODS

The flow of the formative research and intervention development stages are presented in Figure 1.

Figure 1.               Process used

First, a review of the literature was done to identify any potential knowledge gaps in relation to the topic are and the target audience of Jordanian and Syrian families living in Jordan. Second, a mix of individual, family and paired-friendship interviews were conducted. The paired-friendship interviews were only conducted with the children, and the children would select one other friend who they wanted to be interviewed with (i.e., the participants determined who their friends were, as opposed to the researchers or a teacher/parent).

All the interviews were qualitative in nature and were conducted with the aim of generating an in-depth understanding of school-aged children and their families’ nutrition knowledge, attitudes, and practices, including an understanding of the social dynamics around eating preferences and habits within households.

Third, after the findings of the interviews had been analyzed, co-creation workshops were conducted. Co-creation refers to an intervention or campaign design process in which input from consumers plays a central role from beginning to end. The aims of the workshops were to co-design interventions, activities and materials targeted at supporting school-aged child nutrition.

Table 1 details the type and number of participants involved.  

Table 1.          Number of respondents from the different target audiences   

Audience

Number engaged with 

Mothers

89

Fathers

16

Grandmothers

9

Children

6

Individual, family and friendship interviews: Data collection methods and participants   

To help inform and create a structure for the discussions, a guide was initially developed based on the findings from a literature review of articles published on healthy eating and dietary choices. The main topics for discussion were: (a) school-aged children and their families’ nutrition knowledge, attitudes and practices; (b) social and cultural norms that influence children’s nutrition-related behaviors; (c) behavioral determinants of optimal child nutrition, including the barriers and motivators to optimal child nutrition.

The data collection and analysis followed an iterative process, whereby emergent themes from early interviews were used to add to or refine questions during subsequent discussions.  All interviews were conducted during 2018 and 2019 by local external researchers who had training in and experience of conducting qualitative research in Joran. None of the interviews were conducted by WFP staff to avoid potential bias.  The participants were selected using purposive sampling, meaning they were selected because they possessed knowledge that was directly related to the research questions [23]. Sampling considered age, gender, location, nationalities (Jordanian/Syrian), and living situation (living in the community/refugee camp). WFP used a combination of snowball sampling and recruited through local community groups and existing WFP activities, such as the Healthy Kitchen project. The formative data was collected in three locations: Zaatari camp, Ein el Basha, and Karak.  Located in north Jordan, Zaatari is the largest refugee camp in the Middle East and North Africa region currently hosting more than 76,000 Syrian refugees coming from Dara’a district in southern Syria. Zaatari camp and Ein el Basha were chosen as they are vulnerable poverty pockets where WFP undertakes other activities, most notably the school feeding program, and because both Syrian refugees and Jordanians live there. Karak was added to increase geographical spread of the sample. All of the participants had at least one child of primary school age (aged between 5-11 years old) and 25% of the participants were Syrian. 84% were female and the age range of participant was 8 to 71 years old. 

In order for participants to feel at ease during the interview, most of the participants were offered the opportunity to be interviewed in their own home or in a location that was convenient to them. All participants were interviewed once and interviews took between 30 minutes to 1.5 hours, with an average length of 55 minutes. 

Research questions

For this study semi-structured interviews were conducted. At the start of each interview, loosely structured, open-ended questions were asked. In order to pursue an idea or response, more detailed questions were subsequently asked, or prompts made. The wording was not standardised, as the interviewers tried to use the participant’s own vocabulary when framing supplementary questions. 

The questions covered the following areas:

  • current knowledge, attitudes and practices of school-aged children and their families relating to nutrition
  • the knowledge, attitudes and practices of school-aged children and their families’ relating to health and health concerns
  • Social and cultural norms which influence children’s nutrition, including appropriate strategies to influence positive social change and create new social norms
  • barriers and motivators to optimal child nutrition
  • influences school-children’s health and nutrition behaviors
  • appropriate channels, entry points and existing service delivery platforms to reach the target audiences
  • key influencers at the primary, secondary and tertiary levels
  • trusted individuals, organizations, and sources of information

The guide was used as an ‘aide-memoire’ and as a general framework for discussion, ensuring that all themes were covered with the necessary prompts but, at the same time, enabling discussions to be spontaneous, flexible and responsive to the thoughts and opinions of those being interviewed.

Data analysis

All interviews were audio recorded with permission from the participants and transcribed verbatim, then translated into English for analysis purposes. The translations were checked by the national project lead for accuracy before being analyzed. Transcriptions were imported into NVivo [24], and analysis followed a thematic approach to identify key themes and codes [25]. Two researchers from different disciplines analyzed the first five interviews and compared coding. One researcher then went on to analyze the subsequent research data due to the consistency of the initial coding. Data collection and analysis continued until saturation occurred (i.e., the point at which no new significant themes emerged).

No incentives were offered to participants to take part in the interviews.”

Other comments addressed:

  • Line 70: I think the recommendations are for added sugars and suggest a clarification Added in
  • Line 120 and 134: It appears that the main focus of the campaign is to be on child feeding practices but that is not clearly stated in the purpose (line 120) or even the abstract. This focus should be clear in both the abstract and purpose, as well as the key words. Also, it seems that the target audience was mainly from refugees but that is also unclear. Added in
  • Line 121: Co-creation is mentioned but I suggest a clear definition. I figured out later what was meant but think it’s helpful for an international audience to define terms in the intro. Also, what is meant by “choice architecture” (line 84) I have now explained what it is at the start of the methods.
  • Lines 108-119. Fix the formatting here. These are fragments or phrases, not sentences. Also fix lines 100, 105. Typo on line 41  Made changed to lines 108-119 but I am unsure which is the typo being referred to on line 41.
  • Methods section: I have multiple questions/comments here. What was the training or expertise of the staff who conducted the interviews? When transcripts were translated into English, did more than one researcher verify accuracy of the translation? Also, did more than one researcher compare their codes and themes and how were the data merged? This is important to establish validity. Did nutrition program managers, known to the participants, conduct some of the interviews? If yes, what effect might this have had on bias in parent reporting concern about too much sugar intake (for example). I think readers really should be able to see the full questioning guide and suggest the authors make that easily available to readers. In line 125, what was a friendship interview? Also, if families were interviewed, how did that influence response patterns? Were there incentives offered for the interviews? How long did interviews take? All these points have now been detailed / addressed in the methods section.
  • Sample characteristics: Can the authors provide any more details on the characteristics of the sample (beyond Table 1), like educational level of parents, age of children. We have age range (although this was self-reported and the ages of the grandmothers given was rather dubious at times) so I have added in the range as opposed to the average. I also added in a few more details but we did not collect data on their education.
  • Line 309: Define terms “nudge marketing”, “sugar swaps” Added in what we mean and examples
  • Discussion: Compare your findings to those of other researchers who have reported on child feeding practices in immigrant populations. Comparisons now discussed.

Round 2

Reviewer 2 Report

I think the authors have addressed my concerns regarding methodology and discussion of the relevance of their findings to other work. This paper still needs a careful English editing to fix spelling, punctuation, and sentence flow. I also did not see any statement about ethics review of their protocol.